# Using COMPASS (*Context Optimisation Model for Person-Centred Analysis and Systematic Solutions*) Theory to Augment Implementation of Digital Health Solutions

**DOI:** 10.3390/ijerph19127111

**Published:** 2022-06-10

**Authors:** Carey Mather, Helen Almond

**Affiliations:** 1School of Nursing, University of Tasmania, Newnham 7248, Australia; 2Australian Institute of Health Service Management, University of Tasmania, Hobart 7005, Australia; helen.almond@utas.edu.au

**Keywords:** digital, end-user, framework, health, implementation, learning, model, professional, research, social theory

## Abstract

Digital health research is an emerging discipline that requires easy-to-understand theoretical frameworks and implementation models for digital health providers in health and social care settings. The COVID-19 pandemic has heightened the demand for digital health discipline-specific instruction on *how to* manage evidence-based digital health transformation. Access to the use of these models guarantees that digital health providers can investigate phenomena using safe and suitable approaches and methods to conduct research and identify answers to challenges and problems that arise in health and social care settings. The COMPASS theory is designed to aid transformation of health and social care environments. A navigational rose of primary quadrants is divided by four main compass points, with person-centred care being central to the philosophy. Two axes produce Cartesian planes that intersect to form a box plot, which can be used to discover human and physical resource weightings to augment digital health research design and implementation. A third continuum highlights stakeholders’ capabilities, which are critical for any multidisciplinary study. The COMPASS mnemonic guides end users through the process of design, development, implementation, evaluation, and communication of digital health transformations. The theory’s foundations are presented and explained in context of the ‘new normal’ of health and social care delivery.

## 1. Introduction

The increasing complexity of health and social care environments and variation in level of experience, competence, capability, performance expectations, and accountability identified within the health and social care workforce led to the creation of the COMPASS theoretical framework and implementation model. Health and social care now require providers with capability to use digital technology in health and social care delivery. This requirement has been amplified by the global pandemic. For example, there is a renewed emphasis on virtual care, particularly telehealth used in concert with sensors and electronic health records, to provide quality healthcare at a distance while ensuring the safety of the recipients and health and social care workforce. Nevertheless, the acceptance and adoption of digital health and technology frameworks and implementation models continues to be fragmented by research and professions [1]. Digital health is a multidisciplinary set of ideas, notions, and concepts that exist at the intersection of technology and healthcare. To understand this theoretical framework and the implementation model, digital health can be defined as the application and implementation of the digital transformation strategy in the field of health and social care, which includes incorporating both software and hardware solutions as well as services to meet a variety of needs. In summary, digital health is the application of technology (its methods, processes, tools, and devices) to improve individual, family, and community health outcomes while also providing tools to be more aware and make better-informed decisions about health and social care provision [2].

Examples of digital health technologies to understand the scope of the theoretical framework and implementation model include mobile health, health information technology, wearable devices, telehealth, telemedicine, and personalised medicine. From mobile medical apps and software that support clinical decisions made every day to artificial intelligence and machine learning modalities, digital technology is driving a revolution in health and social care. Digital health tools have the potential to improve accuracy in diagnosing and treating illness and disease, as well as to improve individual health and social care delivery. Digital health technologies for health and social care make use of computing platforms, connectivity, software, and sensors. These technologies have a wide range of applications, from general health to medical devices. Digital technologies are also included if they are intended to be used as a healthcare product, in a healthcare product, as companion diagnostics, or as a supplement to other healthcare products such as devices, drugs, and biologics. They can also be used to develop or improve healthcare products [3].

Deliberations regarding how to best engage health and social care providers in digital health technology research in health and social care remain. Furthermore, the literature concentrates on reasons for not adopting digital health and technology: for example, absence of clear accreditation requirements, increased burden on existing health and social care delivery, a lack of capability of the workforce, and financial cost of implementation [4]. There is a need for a mutually agreeable health and social care theory that supports ethical research in digital technology [5]. However, an ethical perspective limited to technology functions is insufficient to assess the broader impact of technology adoption in a person centred, health and social care environment, and the larger health-related ecosystem of which it is a part of [6,7].

In context, the goal of digital health and social care transformation is genuine participatory health [8]. This transformation requires an ecosystem of professionalism, leadership, advocacy, technological acceptance, and behavioural advancement through the creation of trustworthy partnerships of health and social care providers, recipients of care, their significant others, and communities and digital health and social care endeavours to associate digital devices that collect data with the discovery of insights for new models of care [9]. The number and capabilities of digital health solutions continue to expand. Despite these advancements, the confidence of many health and social care stakeholders, ranging from health and social care recipient and providers to payers, industry, and regulators, remains low. As a result, objective, transparent, and evidence-based evaluation of digital health and social care provision is required to provide strong clarity to digital health arenas [10]. One solution is a methodology guided by end-user needs and formal assessment spanning technical, clinical, usability, and cost domains. Quality and value must be easy to differentiate for digital health solutions in order to have a significant impact.

The World Health Organisation [11] proffers a framework for building research environments and encouraging acceptance and adoption of digital health. WHO [11] identifies themes and offers solutions of the following:Evolving study designs that allow an increased implementation of capability innovation;Efficacy and safety, such as comparing digital health and social care options to traditional health and social care methods in well-designed large-scale research;Greater accountability and feasibility motivate researchers to review why some digital health and social care projects have failed;Identifying gaps in planning, dissemination, and enabling the acceptance, adoption, and use of digital health and technologies in health and social care settings is urged by knowledge, attitude, and behaviour;While cost-effectiveness implies that interventions utilising digital health technologies can improve safety and quality of care, long-term cost estimates must be reconciled.

Identifying comparable countries relative to Australia, Canada, the United Kingdom, and New Zealand, the life expectancy is remarkably similar, implying similar lifestyle choices and health awareness as well as access to high-quality health and social care; these counties are linked by similar economic systems, social values, political and legal systems, and the ability to communicate in English [12]. In terms of colonial history, legislation, political institutions, and the socio-economic outcomes of their Indigenous peoples, Australia, Canada, and New Zealand also have much in common [13]. Focusing on digital health, Australia, Canada, England, Scotland, Ireland, and New Zealand have developed their own definitions of digital competence, or capability, and organising domains at national levels (Table 1). To address the lack of digital health acceptance, adoption, competency, or capability, each framework offers a health and/or social care provider-centric framework. Furthermore, although being emphasised as best practice in the adoption and acceptance of digital health, theoretical or research domains or frameworks remain absent.

Each framework (Table 1) was evaluated to determine which digital health capability domains were the most applicable and essential [14,15,16,17,18,19,20,21]. While frameworks differ in terms of focus and terminology, each stress the importance of digital health capabilities delivered within a digital health ecosystem and considered innovation, communication, collaboration, participation or engagement, digital identity, and satisfaction in the development of digital skills among workforces. The guidance these digital capability frameworks provide to governments, organisations, and individuals in recognising the repercussions of the digital revolution’s developments is one of their most important contributions. However, none of the investigated frameworks established a health and social care domain specifically for *how to* research digital health solutions. The authors researched and developed COMPASS, a theoretical framework and implementation model for improving the deployment of digital health solutions and capability in health and social care settings to overcome this deficit.

The theoretical framework and implementation methodology provides a platform for presenting a transformative digital health philosophy. COMPASS is a visual and narrative theoretical framework and implementation model. The COMPASS mnemonic can assist digital and social care providers in approaching any study in a systematic, person-centred manner. COMPASS encompasses variables digital health and social care providers must, prior to embarking on any activity, address digital health concerns while planning, producing, executing, and analysing any digital health research or quality assurance process. COMPASS is a platform that guides individuals or teams through the process of conducting digital health research.

## 2. Materials and Methods

A staged approach was employed to construct the COMPASS theoretical framework and implementation model. A search for recently (2018–2021) published digital health research on competency and capability strategies and frameworks was conducted, and information was systematically synthesised. The theoretical framework was refined over a 12-month period through multiple iterations, beginning with a literature and expert review. To ensure rigour, 33 experts from an Australasian digital health community of practise (CoP) were invited to participate in a cloud-based focus group in July 2021. CoP was represented by twenty-four members. The visual images and accompanying narrative were distributed to participants. The authors presented the theory and then solicited feedback through open-ended questions about each layer and explanation. The revised theoretical framework and implementation model was emailed to participants for individual feedback. In August 2021, a further validation meeting of CoP participants was held to ensure that the authors understood expert suggestions and validated any changes on the revised iteration. The most recent version was refined further after considering the CoP input. The development procedure is described. No ethics approval was required, all in information collected is available in the public domain.

### Review of Evidence

A preliminary assessment of published research and grey literature on digital health competency and capability models and frameworks was undertaken during April to September 2021. The primary focus of the searches was to locate any published government or professional regulatory authority information related to digital health competency, capability or implementation of digital health strategies or solutions within jurisdictions. Further searches of published and grey literature via Google Scholar and library catalogue based mega searches that included PubMed, Scopus, ProQuest, and CINAHL were also undertaken using search terms such as digital, health, capability, competency, ecosystem, framework, and strategy. The search discovered a corpus of research on health and social care provider digital health capabilities [14,15,16,17,18,19,20,21]. However, the competencies and capabilities discovered were limited to specific types of healthcare providers, and none of the solutions provided guidance for *how to* prepare future health, social care, or digital students and educators regarding the implementation of digital health research solutions.

A review and synthesis of evidence revealed commonalities that allowed the COMPASS theoretical framework and implementation model to be developed. In the first instance, four quadrants were agreed upon, and these serve as the theoretical framework’s foundation. To support health and care, leadership and advocacy are required to ensure governance systems support individual, organisational, community, national, and international levels of health social care and service provision (Leadership and Advocacy); The positive influence of technology on health and social care must be articulated in transformative digital health care (Transformative Care); when utilising digital media, professionals must comprehend, develop, and know acceptable professional behaviour (Digital Professionalism). Informatics and quality must be synonymous since digital health and social care depends on appropriate acquisition, recording, and storage of high-quality data (Data Information and Quality).

## 3. COMPASS Theoretical Framework and Implementation Model

*Context Optimisation Model for Person-centred Analysis and Systematic Solutions* (COMPASS) in digital health is a theoretical framework and implementation model that can be used by digital health and social care providers to augment the transformation of health and social care environments. When end users are planning to research or investigate phenomena in health and social care environments, COMPASS and the mnemonic provide guidance on *how to* identify relevant research methodologies. By using quadrants and three continuums, end users are provided with directions on *how to* choose and utilise relevant research methods to address digital health concerns. COMPASS focuses on person-centred care, which places health and social care recipients at the centre of the theoretical framework. The ability to embed technology and improve health outcomes can be achieved by using an authentic participatory health strategy. The COMPASS mnemonic augments digital health research by giving health and social care providers tangible direction on *how to* plan, develop, implement, and evaluate digital health solutions in health and social care environments.

The aims of the COMPASS theoretical framework and implementation model are as follows:The navigational compass rose is the inspiration for COMPASS. It is a visual representation of a theoretical framework and implementation model that shows *how to* plan, develop, implement, and evaluate digital health solutions for digital health and social care providers. The COMPASS theoretical framework contributes to digital health transformation by enabling identification of the foci of digital health issues including human and physical resources and the identification of capability of stakeholders.The COMPASS mnemonic summarises the steps involved in *how to* conduct digital health research. Each letter of the mnemonic represents a step in the process of planning, developing, implementing, and evaluating digital health research. By ensuring that components of the research process are included in each suggested solution, the mnemonic aids digital health transformation.

### 3.1. Overview

A core, quadrants, two layers, and three continuums form the COMPASS theoretical framework and implementation model. COMPASS is also a mnemonic for the research process that digital health and social care providers can use to guide their efforts. The navigational rose core is person-centred care, which is the focus of digital health transformation (Figure 1).

The person remains at the centre irrespective of the distance from the end user and is the recipient of care. For example, a recent employee in a government digital health department had the role of clinical governance implementation manager. Despite not being directly involved in the delivery of person-centred clinical care, their goal remains to be directed towards ensures appropriate health-outcomes for health or social care recipients. The primary quadrants form the first or base layer of the COMPASS, providing foundational direction to researchers when considering undertaking any digital health or social care transformation (Figure 1).

Leadership and Advocacy, Transformative Care, Digital Professionalism, and Data Informatics and Quality form the quadrants (Figure 2). Four main compass points, Human, Technical, Clinical, and Wellness, provide end users with direction, forming two continuums (Figure 3).

The four COMPASS points become a Cartesian plane or scatter box plot graph tool that can be used to guide human, technical, clinical, and wellness COMPASS point components and enable researchers to understand where on the continuum or within the box plot are digital health gaps or opportunities are focused towards. The vertical and horizontal axes of the Cartesian plane provide visual cues for thinking about the first continuum of human centeredness and technology, as well as whether the solution is focused on clinical or wellness. The graphing tool can be used by researchers to plot where the suggested digital health solution fits on the continuum’s axes. Within the context of these directions, a weighing of the human and physical resources required will be disclosed.

A capability continuum is the third continuum (Figure 4). Digital health and social care providers can utilise this continuum to assess their capability, as well as identify the capabilities and priorities of other stakeholders. There are three levels of capability in this continuum: empowered, transitional, and entrusted. Providers beginning their digital health journey are empowered towards developing their capability. As their knowledge, skills, and capacity grow, it is expected that these digital health and social care providers would become transitional in capability. The final stage of this continuum is entrusted as digital health professionals become independent for conducting and leading research in the field.

### 3.2. Layer 1: Core and Quadrants

#### 3.2.1. Person-Centred Care

Person-centred care refers to individuals receiving care at the centre of any digital health solution. Information must be presented in a format, language, and terminology that recipients of care or their significant others can comprehend and relate to, allowing them to make informed decisions and exert control over their life [22].

#### 3.2.2. Leadership and Advocacy

Providers of health and social care are critical to the transformation of health care. Individual, organisational, and national leadership and advocacy are required to influence governance and decision making to ensure quality and safety and to enhance health outcomes. Digital health and social care providers can drive research planning, development, implementation, and the evaluation of digital health technologies in health and social care environments through leadership and advocacy.

#### 3.2.3. Transformative Care

Transformative care is the process of using digital health technologies to have a positive impact on health and social care environments. It involves end users engaging, navigating, and interacting with care recipients to share data and make decisions based on digital communication that can improve health outcomes [23].

#### 3.2.4. Digital Professionalism

Digital professionalism is a component of professional identity, and it embodies the values of the various health and social care providers. The term digital professionalism describes the need to “understand, develop and know appropriate professional behaviour when using digital media” ([24], p. 1).

#### 3.2.5. Data Informatics and Quality

Privacy, security, and confidentiality are critical for assuring reliable data informatics and quality. For high-quality, accurate, systematic, and timely data collection, an understanding of information generation, application, and management is required.

### 3.3. The Main Compass Points

The four primary compass points that comprise the COMPASS continuum form a vertical and horizontal axis. On each plane, these points symbolise the transition from human towards technical and clinical towards wellness components. Each forms the Cartesian plane that intersects to generate a box plot, which can be used to identify and quantify values associated with the continuums’ quadrants (Figure 2). The box plot’s purpose is to offer health and social care providers a visual representation of any proposed digital health solution’s focus. Researchers can understand the focus of the proposed project by reviewing inside the two continuums where the plot is in the value quadrants.

### 3.4. COMPASS Directions Continuums

#### 3.4.1. Layer 2a: Human and Technical Continuum

In health and social care settings, both human and technical aspects are ubiquitous. The box plot can be used to visualise the burden of human and physical resources, which can assist in understanding human and technical aspects of any proposed digital health solution. If human and technical complexity is not considered prior to planning, development, implementation, or evaluation of any proposed digital health solution, workarounds or unforeseen consequences may arise [25].

#### 3.4.2. Layer 2b: Clinical and Wellness Continuum

In the context of this continuum, clinical and wellness refer to the many degrees of health care required to maintain and improve one’s health and wellbeing (Figure 3). Recipients of care with more complex needs would be plotted closer to the clinical end of the continuum, whereas an ambulant individual in a community would be plotted closer to the wellness end. Individuals may require a variety of technological interventions. For example, ambulant individuals in a community are more likely to use technology for wellness monitoring and maintenance. The recipients of care in tertiary health facilities are more likely to require more complex care that includes the use of more specialised digital health technologies.

The weightings of each of the four components of the key COMPASS points can be estimated using the two continuums. The value of plots can provide direction to digital health and social care providers, allowing for the estimation of human and physical resources. The other minor COMPASS points have been kept blank on purpose. These locations can be filled in by researchers, allowing them to pinpoint and refine their focus. It allows stakeholders to view if there is more weighting in any of the quadrants. This graphic representation provides further information to guide planning or developing any proposed digital health solutions.

#### 3.4.3. Layer 3: Capability Continuum

The capability continuum is the third of the COMPASS theoretical framework and implementation model’s three continuums, and it overlays the quadrants (Figure 5). Knowledge, skills, abilities, and behaviours in digital health are developed over time by health and social care providers through experience. This development is shown visually by the amplification of colour that radiates outward from the core and visually symbolises capability developments. The lighter colour at the centre of the navigational rose represents beginning providers commencing their digital health journey, empowered to pursue digital health solutions. As digital health providers improve their knowledge, skills, and capabilities, they will transition towards undertaking digital health research. Finally, digital health and social care providers will be entrusted with the responsibility of leading and advocating for digital health solutions in health and social care environments as experts, represented by the increased depth of colour. Table 2 provides a narrative description of third continuum of the transition in knowledge, skills, behaviour, and ability from empowered to entrusted capabilities.

##### Empowered

Providers of health and social care are given the opportunity to participate in and learn about digital health and social care research. To become transitional in capacity, providers must have the opportunity to be exposed to digital health challenges to widen their knowledge, skills, abilities, and behaviours. The move from empowered to transitional capability is gradual, and it is a learning process that occurs over time and through exposure, allowing for the growth of capability through experience learning.

##### Transitional

After developing knowledge, skills, abilities, and behaviours, digital health and social care providers acquire an intermediate level of expertise in understanding, knowing, and leading digital health research. During this period of change, digital health and social care providers may continue to build capacity in some COMPASS directions while becoming experts and entrusted in others. At this point, digital health and social care providers may start mentoring, coaching, and leading research into digital health solutions.

##### Entrusted

Entrusted providers of digital health and social care have acquired a level of proficiency that allows them to lead and supervise research while also mentoring and training others in digital health solutions [26]. Entrusted denotes an elevated level of capability, indicating that digital health and social care providers have become experts in digital health decision making and have earned the trust of peers, colleagues, and stakeholders to lead digital health research.

#### 3.4.4. How to Use the Capability Continuum

The capability continuum’s purpose is to encourage health and social care providers in determining their own capabilities as well as the capabilities of others. The objective of the continuum is to allow for the identification of stakeholders’ levels of capability to ensure that any interdisciplinary digital health team has the appropriate balance of knowledge, skills, and capabilities, which are critical for the success of digital health research. An unbalanced representation of foundation quadrant capability within a team might result in unintended consequences or the failure of digital health solutions.

### 3.5. COMPASS Mnemonic

#### 3.5.1. Aim and Purpose

The theoretical framework and implementation models are supported by the COMPASS mnemonic. The letters of the word COMPASS form the beginning of a mnemonic statement that represents the process of *how to* consider or approach any potential digital health solution. The meaning of each letter in the term COMPASS allows researchers to apply a digital health perspective to practical considerations while planning, developing, implementing, and evaluating digital health solutions.

The COMPASS mnemonic statements consider the following:Context—circumstance, situation, place, or time where digital transformation can influence health and social care safety, and quality;Optimisation—methodologies adopted to ensure evidence-based therapeutic interventions that influence effective use of digital technologies to prevent, manage, or treat health disorders or diseases;Model—plans, designs, implementations, and evaluations which best provide representations of digital technology within health and social care, learning approaches that contribute to safety and quality;Person-centred—values, holistic lens and inclusive goals facilitated by digital technology-enabled care, giving individuals authority to better engage with and control their health;Analysis—rigour, reliability, and validity ensuring detailed examination of digital transformations in health and social care to understand the nature or determine essential features;Systematic—methods required to develop and deliver a transformative approach in digital health and social care that is logical, repeatable, and able to be learned as an organised approach;Solution—research impact, transformation, and future directions. The solution resolves a concern using a transformative approach in digital health and social care.

#### 3.5.2. How to Use the COMPASS Mnemonic

The COMPASS mnemonic provides researchers in health and social care practical guidance on *how to* design digital health solutions. It is intended to augment research processes rather than provide any specific approach or methodologies. The COMPASS mnemonic can be used to address small and modest, as well as large and complex digital health or social care challenges. The COMPASS mnemonic is useful in a variety of settings, including organisational, administrative, clinical, educational, technical, and research. The mnemonic will become embedded in digital health and social care provider research practice as end-users develop knowledge, skills, experience, and capability. The mnemonic is likely to be used to lead or steer the creation of digital health and social care solutions by empowered and transitional levels of capacity. Those researchers who have achieved entrusted capability will have embedded the COMPASS mnemonic into their knowledge, skills, abilities, and behaviours.

## 4. Discussion

### Why the COMPASS Is Different

While the digital health capability frameworks interrogated specify what content should be included, there are currently no practical discussions or domains offering digital health and social care relative to *how to* research solutions. More information about development techniques that health and social care providers might utilise to remain contemporary in digital health and social care provision has yet to be thoroughly explored in the ‘new norm’ of healthcare provision.

Furthermore, a significant shift from traditional and single-disciplinary academic and clinical approaches is required for the successful development, integration, and deployment of digital health, technology, and methodologies. There is a need for innovative approaches to science, health, and social care research to appropriately embrace, reform health and social care, and improve health outcomes. Only when health, social care, and technology are researched synergistically and focused on solving real-world issues will these fields be able to provide discoveries that have a major influence on clinical and social care delivery and improve the wellbeing of individuals and populations.

The COMPASS theoretical framework and implementation model was developed to augment implementation of digital health solutions and capabilities in health and social care environments. COMPASS provides digital health and social care researchers with a theoretical framework and implementation model that supports appropriate known research processes, methodologies, and methods. The two layers consisting of the foundational core and quadrants, and the second layer consisting of four main compass points creating the Cartesian plane of two continuums forming a box plot that can be used to identify and clarify the foci of human and physical resources required to enable digital health researchers to investigate digital health issues. It is critical to augment digital health and social care providers ability to plan, develop, implement, and evaluate digital health solutions in health and social care environments. The capability continuum allows end users to assess their own and others’ capabilities, which is critical for the development of any research team.

The acceleration and rapid uptake of digital technology in response to circumstances created by the COVID-19 pandemic has indicated the need to establish evidence-based research theory and implementation models for digital health and social care providers. The evaluation of the COMPASS is required to determine the levels of success, measured in a range of factors including health outcomes, cost-effectiveness, or efficacy. Other lenses such as quality and safety need also to be evaluated to understand which strategies need to be continued and those that are no longer required, are inefficient, or ceased due to other reasons such as workarounds that have unintended negative consequences or are unsustainable [23]. Technology continues to rapidly evolve and monitoring and evaluation are necessary to ensure that the most effective, safe, and high-quality care is provided in the ‘new norm’ of health service delivery.

## 5. Limitations

Digital technology is evolving rapidly, and digital health transformation needs to keep pace. There is lag time between when direction is needed and publication of government policy, which hinders innovation and legitimate uptakes by health providers within health and social care environments. Theories and implementation frameworks need to be robust however, they are only useful when perceived as relevant behaviour modification theories. Bandura [27] and Prochaska and DiClemente [28] indicated that self-efficacy, action, and maintenance behaviours require optimal conditions before acceptance and integration into the ‘new normal’ occurs. The purpose of the COMPASS is for end-users to be able to deploy the COMPASS theory and implementation model and mnemonic to understand *how to* augment digital health and technology transdisciplinary approaches to behaviour modification research in terms of whether stakeholders identify as empowered, transitional, or entrusted health and social care providers. The COMPASS theoretical framework and implementation model was developed as a contemporary approach within the ‘new norm’ of health and social care provision. However, it will need to be updated to reflect change within this growing discipline.

## 6. Future Directions

Feedback on the use, usefulness, and utility of the COMPASS theoretical framework and implementation model will assist in enhancing and improving the theory and implementation approach. The COVID-19 pandemic has accelerated the uptake and application of digital health technology in health and social situations, allowing many critical lenses to guarantee that COMPASS is a contemporary approach to digital health research. Support for enabling digital health solutions will remain critical in the ‘new normal’ of health and social care delivery.

## 7. Conclusions

The COMPASS theory and implementation model were created in visual and narrative formats to aid digital health and social care providers in navigating, exploring, and comprehending *how to* research and solve digital health and social care concerns. An analysis of current digital health professional capability frameworks and ecosystems from countries with similar health systems was undertaken. A synthesis of these frameworks and models resulted in the development of the COMPASS theoretical framework and implementation model. The COMPASS mnemonic supports digital health research by guiding end-users through the process of establishing human and physical resources and stakeholder capabilities required to guide the planning of potential digital health transformation solutions in the context of the ‘new norm’ of health and social care provision.

## Figures and Tables

**Figure 1 ijerph-19-07111-f001:**
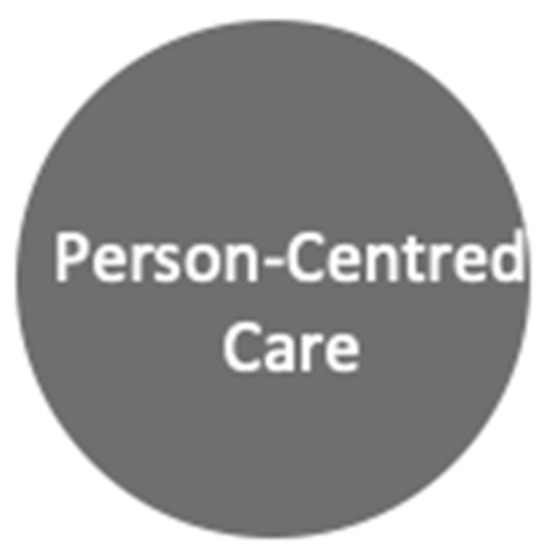
Core of COMPASS person-centred care.

**Figure 2 ijerph-19-07111-f002:**
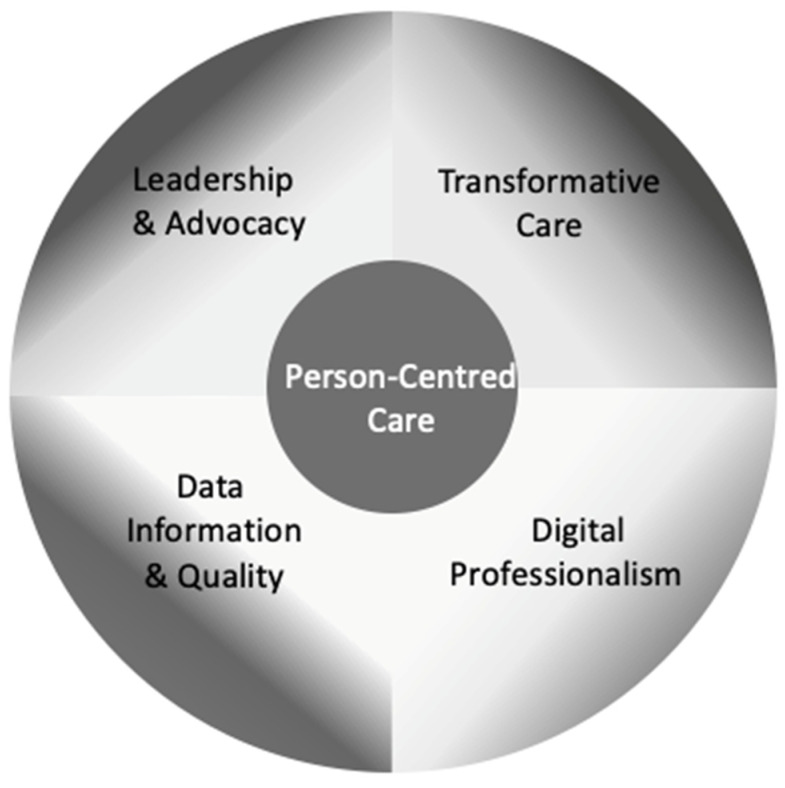
Quadrants of COMPASS theoretical framework and implementation model.

**Figure 3 ijerph-19-07111-f003:**
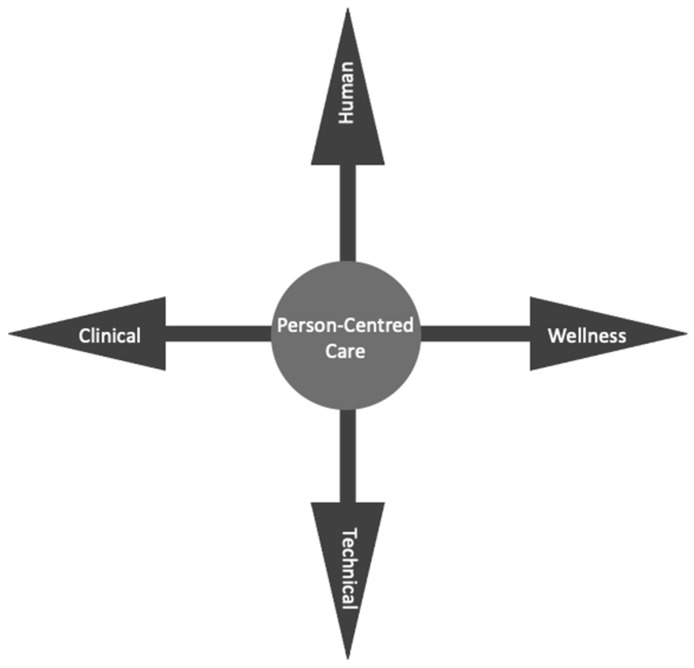
Layer 2a: Human and technical continuum (vertical axis). Layer 2b: Clinical and wellness continuum (horizontal axis) form a Cartesian plane scatter box plot graphing tool.

**Figure 4 ijerph-19-07111-f004:**
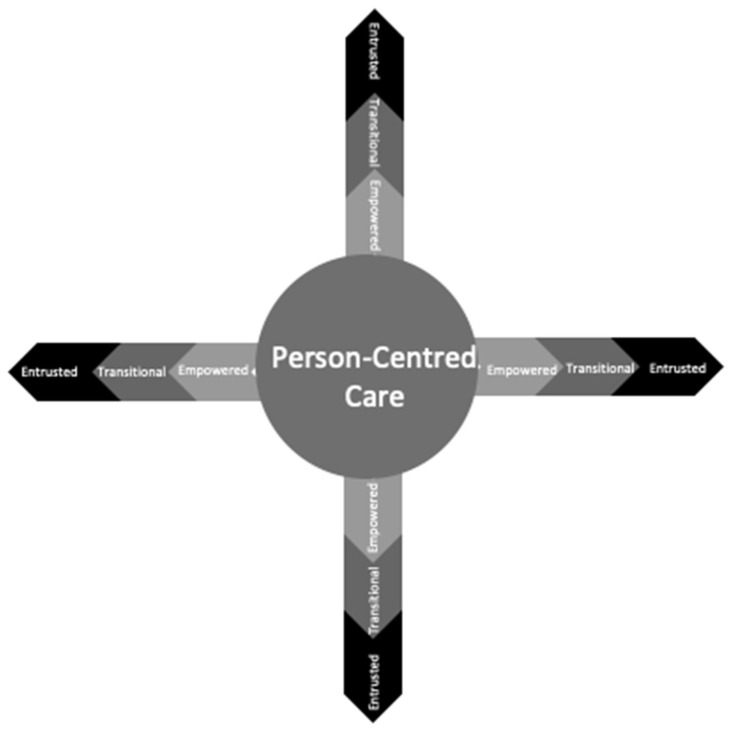
Layer 3, Capability continuum.

**Figure 5 ijerph-19-07111-f005:**
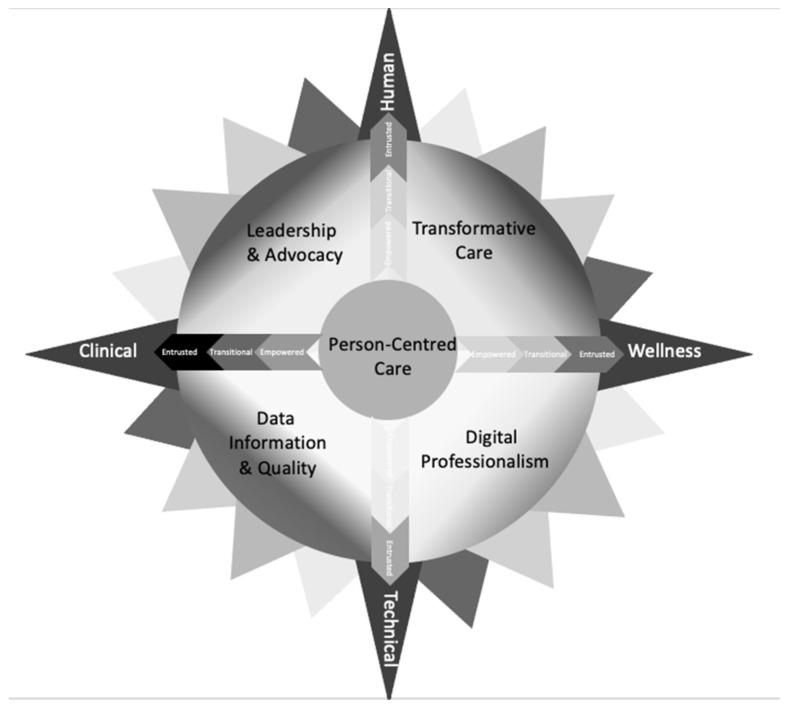
Visual representation of COMPASS theoretical framework and implementation model.

**Table 1 ijerph-19-07111-t001:** The key domains of current digital capability frameworks.

Country	Title	Profession	Core Purpose	Domain 1	Domain 2	Domain 3	Domain 4	Domain 5	Domain 6	Domain 7
Australia 2020 [14]	National Nursing and Midwifery Digital Health Capability Framework	Nursing and Midwifery	Person/woman-centred, safe, quality and connected care	Digital Professionalism	Leadership and Advocacy	Data and Information Quality	Information-enable care	Technology		
Australia 2021 [15]	A Capability Framework in Digital Health in Medicine	Medical education	Culturally safe, people, and value-based care	Professionalism and Interagency Action	Integrated health settings and access	Appraisal and risk	Data and information Quality	Medicine Ethics and law	Future Preparedness	Health System Innovation
Australia 2021 [16]	Digital health capability framework for allied health professionals	Allied Health	Safe and high-quality patient care	The digital workplace	Digital Professionalism	Data and informatics	Digital transformation			
Canada 2019 [17]	Digital Health Canada Competency Requirements	Health informatician		Information Management	Technology Eco System	Clinical and Health Services	Canadian Health System	Healthcare Transformation	Project Management	
England 2018 [18]	A Health and Care Digital Capabilities Framework	Health and Care	Person Centre Digital Literacy	Information, data, and content	Teaching and learning and self-development	Communication, collaboration, and participation	Digital identity, wellbeing, safety, and security	Technical proficiency	Creation and innovation and research	
Scotland 2020 [19]	Digital Capability: A Scottish Landscape Review	Lecturers		Identifying staff digital skill needs	Who needs what at which level	The development of those skills	Implementing a data-driven approach to learning and teaching	Ensuring an effective and resilient digital infrastructure		
Ireland 2021 [20]	All-Ireland Nursing and Midwifery Digital Health Capability Framework	Nursing and Midwifery		Digital professionalism	Leadership and Advocacy	Data Information and Quality	Information-enable care	Technology		
New Zealand (n.d.) [21]	Digital Health strategic Framework	Public health and disability system	Person and whanau centred. Customer driven	Delivering and improving digital health services ecosystem	Enabling the ecosystem	Digital Environment	Digital and data delivery principles			

**Table 2 ijerph-19-07111-t002:** COMPASS Capability continuum.

	Empowered	Transitional	Entrusted
Knowledge	Foundational	Proficient	Advanced practice
Skills	Able	Discern	Competent/Capable
Behaviour/Ability	Receptive	Motivated	Change champion/Leader

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
