# Peer review of "Using COMPASS (Context Optimisation Model for Person-Centred Analysis and Systematic Solutions) Theory to Augment Implementation of Digital Health Solutions"

_ijerph, 2022, doi:10.3390/ijerph19127111_

Round 1
Reviewer 1 Report
The authors here demonstrated the design, creation and implementation of the COMPASS theory in visual and narrative format, to aid digital health and social care providers in navigating, exploring, and
comprehending how to research and solve digital health and social care concerns. Digitization has proved its efficiency in improving social and healthcare services and hence the entire healthcare industry is investing heavily on digitization. The focus has just increased multiple times at the post-Covid-19 pandemic recovery phase. In this context, the current work has considerable importance to the field. The authors have discussed the background of the work in a compact but apt manner and then elaborated the idea step by step. Both a descriptive and visual analysis were well discussed, and the methods and limitations were properly discussed. Hence I would recommend the article to be published in its present form, after a regular English grammar and spell check.
Reviewer 2 Report
The manuscript is original, interesting, useful for potential readers, and well-illustrated. The author's idea is clear and explained in detail, and the conclusions are applicable in research practice. The text would gain quality if the authors briefly commented on how this model fits into the nine core principles of digital health ethics (https://www.amrc.org.uk/).
A preprint version of the paper has already been available since May 4, 2022, at: https://www.preprints.org/manuscript/202205.0002/v1/download. How do you explain that?
Reviewer 3 Report
1 - Material and Methods: 22 experts - how was the interview conducted? Focus group, open questionnaire, closed questionnaire? This information is essential when thinking about replicating the work in other regions or countries
2 - Review of evidence: In this section, do the authors mention that the period analyzed was only from April to September 2021? I was in doubt: Was this period the delimitation of the research, or was it the time they took to analyze the current literature? It was unclear.
3 - The article talks about "digital health," but what is digital health? I think there should be a specific section talking about this.
4 - In section 3.1.3, the authors talk about "digital health technologies." What technologies exist that are classified as digital health Technologies?
5 - Finally, the Framework is based on empirical studies, and the authors themselves say so. The Framework shows several criteria that should be considered to implement digital health technologies. As it is an empirical study, I see that it should be listed the digital health technologies that appear in the literature studied and how the Framework can help implement each one.
Round 2
Reviewer 3 Report
Thank you for considered all my suggestions.